# Preparation of Polyvinylidene Fluoride Nano-Filtration Membranes Modified with Functionalized Graphene Oxide for Textile Dye Removal

**DOI:** 10.3390/membranes12020224

**Published:** 2022-02-15

**Authors:** Hirra Ahmad, Muhammad Zahid, Zulfiqar Ahmad Rehan, Anum Rashid, Saba Akram, Meshari M. H. Aljohani, Syed Khalid Mustafa, Tayyaba Khalid, Nader R. Abdelsalam, Rehab Y. Ghareeb, Mohammad S. AL-Harbi

**Affiliations:** 1Department of Chemistry, University of Agriculture, Faisalabad 38000, Pakistan; hirraahmad25@gmail.com (H.A.); rmzahid@uaf.edu.pk (M.Z.); tayyabakhalid59@gmail.com (T.K.); 2Department of Materials, National Textile University, Faisalabad 37610, Pakistan; anumrashid800@yahoo.com (A.R.); saba.akram1980@gmail.com (S.A.); 3Department of Chemistry, Faculty of Science, University of Tabuk, Tabuk 71491, Saudi Arabia; mualjohani@ut.edu.sa (M.M.H.A.); khalid.mustafa938@gmail.com (S.K.M.); 4Agricultural Botany Department, Faculty of Agriculture (Saba Basha), Alexandria University, Alexandria 21531, Egypt; nader.wheat@alexu.edu.eg; 5Plant Protection and Biomolecular Diagnosis Department, Arid Lands Cultivation Research Institute, The City of Scientific Research and Technological Applications, New Borg El Arab, Alexandria 21934, Egypt; ryassin@srtacity.sci.eg; 6Department of Biology, College of Science, Taif University, P.O. Box 11099, Taif 21944, Saudi Arabia; mharbi@tu.edu.sa

**Keywords:** membranes, nano-filtration, phase inversion, biofouling, sustainability

## Abstract

Water scarcity has become one of the most significant problems globally. Membrane technology has gained considerable attention in water treatment technologies. Polymeric nanocomposite membranes are based on several properties, with enhanced water flux, high hydrophilicity and anti-biofouling behavior, improving the membrane performance, flexibility, cost-effectiveness and excellent separation properties. In this study, aminated graphene oxide (NH_2_-GO)-based PVDF membranes were fabricated using a phase-inversion method for textile dye removal. These fabricated membranes showed the highest water flux at about 170.2 (J/L.h^−1^.m^−2^) and 98.2% BSA rejection. Moreover, these membranes removed about 96.6% and 88.5% of methylene blue and methyl orange, respectively. Aminated graphene oxide-based polyvinylidene fluoride (PVDF) membranes emerge as a good membrane material that enhances the membrane performance.

## 1. Introduction

Freshwater scarcity has become a threat to the sustainable development of human society in the last few decades [1]. The essence of global water scarcity is the geographic and temporal mismatch between freshwater demand and availability [2,3]. There are different sources of water pollution, which consist of agricultural, municipal and industrial wastes. Wastewater includes microorganisms, pesticides, fertilizers, halogenated compounds, organic acids, toxic heavy metals, textile dyes, etc. Moreover, various diseases are caused by the intake of polluted water; these include skin ulcers, irritation, diarrhea, headache, appetite loss, fever and abdominal pain. The increasing consumption of industrial textile dyes has been caused by major environmental problems issued from industrial activities. Textile dyes are often complex organic molecules, which are usually difficult to break down biologically. They are to be attached to fabric fibers to confer color to them. If these more common organic compounds are discharged into water bodies, they may mean real sources of pollution. Therefore, to ensure a clean water supply, it is necessary to remove pollutants from wastewater.

There are various techniques for the treatment of wastewater. Membrane technology plays a role in solving the major worldwide crisis of water [4]. Membrane performance is the major factor for the development of membrane separation technology. Among all the techniques for this purpose, nanofiltration technique plays the largest role. This technology is generally used in wastewater treatment methods because of its low-cost and effective process [5,6,7]. Generally, most polymeric nanofiltration membranes possess flexibility and low cost; however, they have some major issues, such as poor chemical resistance, restricted life period, and fouling of the membrane [8]. 

Polymeric membranes are becoming more economical as compared to other membranes because of their flexibility, ability to easily roll up into spiral wound modules or hollow fibers, and their solution processability [9]. Polymeric membranes have low resistance to high temperature and are the most widely used membranes for the treatment of wastewater [10]. Polymeric membranes have easy-forming properties and selective transfer of chemical species and are less expensive than other membranes. Polyvinylidene fluoride (PVDF) is a semi-crystalline polymer that contains amorphous phase and crystalline phase. The amorphous phase offers flexibility approaches to membranes, whereas the crystalline phase provides thermal stability [11]. PVDF is hydrophobic in nature and has very good resistance. Polyvinylidene fluoride membranes are widely used in ultrafiltration, membrane bioreactors and microfiltration. These types of membranes are mostly prepared by the phase inversion method because of their simplicity. The phase inversion method is frequently used because of its effectiveness in preparing polymeric membranes. The phase inversion method is a versatile technique and is widely used to prepare symmetric and asymmetric membranes because of its immersion precipitation [12]. This method has been employed to fabricate membranes. The incorporation of polyurethane in blended membrane comprising of polymers of (polyether sulfone/ polyether urethane (PES/ETPU) results tremendous improvement in surface morphology, hydrophilicity as well as porosity and pore size distribution. It is known that with an increase in concentration of additive, increased viscosity is observed, which acts as a barrier in order to transfer the mass between solvent and non-solvent. Thus, a delay rate of demixing reduces the pore size and permeability [13].

A major limitation of membrane technology is membrane fouling. This is a phenomenon in which undesirable species, either living organisms or non-living substances, attach to the surface of the material, either internally or externally. Biofouling is considered a biotic form of organic fouling and is the most common among them. It mainly occurs in microfiltration, ultrafiltration, nanofiltration, and reverse osmosis [14]. Biofouling takes place through the addition of disinfectants such as chlorine or the use of pretreatment systems. Bacteria that are implanted in the biofilm are unaffected by biocides, as compared to the same bacteria present in the dispersed state. Therefore, bacteria are related to several contaminated substances that are generally used for separation in membrane processes [15]. In order to avoid membrane biofouling, fabrication of membranes can be performed. Fabrication enhances the water permeability and membrane selectivity. However, it is difficult because it decreases the selectivity of the membranes that are present within the porous structure. Fabrication enhances the characteristics of membrane like surface smoothness, hydrophilicity, surface charge, and its antibacterial behavior [16]. Membrane fabrication can be done via blending of nanoparticles within membranes and the entrapped nanoparticles on the surface of membrane. Nanomaterials show greater antibacterial properties, which are chemically inert but weak oxidants in water. Nanomaterials have very good properties such as high surface area, high reactivity and a very small size that make them behave similarly to sensors, catalysts and adsorbents [17]. 

Graphene is an sp^2^ hybridized carbon network that has gained much attention due to its mechanical, thermal and thermoelectric properties [18]. These properties of graphene make it a very beneficial for numerous uses in the field of memory devices, anti-corrosion coatings, oxygen reduction reaction and superconductor devices [19]. Graphene oxide enhances membrane performance, and functionalization of graphene oxide is performed to obtain efficient performance and properties. The compatibility and properties of the polymer and graphene oxide are enhanced by introducing different functional groups present on the graphene oxide surface. The amino group has good chemistry and high reactivity with the polymer. Nitrogen is a heteroatom that can be introduced on carbon nanomaterials through chemical vapor deposition (CVD) or thermal treatment [20]. The purpose of the current study is to synthesize functionalized graphene oxide embedded in the polymer matrix to fabricate polymeric nanocomposite membrane for textile dye removal from industrial wastewater. The permeability, water flux and stability of the membrane was checked, and the anti-biofouling property of membranes was assessed through bovine serum albumin (BSA) rejection.

## 2. Materials and Methods

### 2.1. Reagents, Materials Preparation and Fabrication

Graphite, sulfuric acid and sodium nitrate from Merck, Germany, was used. Ethylene glycol, ammonia, hydrogen peroxide, polyvinylidene fluoride and potassium permanganate were obtained from DAEJUNG, Korea. All aqueous solutions were prepared in distilled water.

Graphene oxide was synthesized using a modified Hummer’s method [21]. Initially, the preparation of graphite oxide was performed with the addition of 5 g graphite powder, 2.5 g sodium nitrate (NaNO_3_) and 200 mL sulfuric acid (H_2_SO_4_) into the flask with continuous stirring. Later, 30 g potassium permanganate was slowly added into the suspension-containing flask and kept in an ice bath in order to maintain a temperature of 5 °C. This process was continuously maintained for 3 h. The temperature was raised to about 35 °C for 45 min after the ice bath was removed. The temperature of the suspension was raised to 98 °C for 20 min, and approximately 200 mL distilled water was added into it. The suspension color turned to brown. Further, the suspension was diluted with 250 mL distilled water, and then 20 mL of 10% hydrogen peroxide was added in order to reduce the unreacted manganese dioxide and permanganate to soluble manganese sulfate. The color of the suspension turned bright yellow from the addition of hydrogen peroxide. The suspension was centrifuged, resulting in yellowish-brown thick paste. With the help of centrifuge, this paste was washed three times with 95% alcohol and deionized water, resulting in graphene oxide. Thus, this graphite oxide was sonicated for the separation of layers of graphene oxide. Then, resulted graphene oxide was dried in oven at 60 °C for 12 h. The schematic representation of synthesis of graphene oxide in shown in Figure 1.

The aminated graphene oxide was synthesized by the solvothermal reaction with graphene oxide and ethylene glycol in an autoclave reactor. First, 1 g of graphene oxide was sonicated in 25 mL of distilled water that was further added into 210 mL of ethylene glycol in a beaker. After this step, 6 mL of ammonia water was added into it and placed in the autoclave for chemical reaction under high temperature and pressure for 15 h. The resultant precipitates were strained and wash away through distilled water until the neutral pH was attained. Then, the resultant mixture was dried out in the oven at 60 °C for 7 h [22]. The schematic representation of synthesis of aminated graphene oxide is shown in Figure 2.

Aminated graphene oxide–based polyvinylidene fluoride membranes were fabricated through the process of phase inversion. Polyvinylidene fluoride polymer with 18% different concentration of aminated graphene oxide was added to dimethylacetamide solvent for the casting solution preparation. The homogeneous solution was obtained with continuous stirring for 4–6 h. For the removal of air bubbles, the solution was left overnight. After that, the casting solution was cast over the casting machine. After 90 s, the membrane was placed in a coagulation bath for 2–3 min. After the detachment of membrane from the surface, it was washed three times with pure water. Then, the membrane was dried at room temperature [23]. The compositions of different polyvinylidene fluoride membranes are presented in Table 1.

### 2.2. Membrane Performance

All the experiments were performed in a filtration assembly with an area of the membrane of 50 cm^2^. The membrane was immersed in ultra-pure water for 3 h and then compacted at 5 bar transmembrane pressure (TMP) for 1 h. Permeability measurements were conducted at room temperature, and the pressure was varied from 1 to 4 bar. Throughout the filtration process, using dye solutions, the permeate samples were collected to make a certain concentration of the feed solution. Afterwards, the membrane was washed using distilled water, and the fouled membrane permeability was measured with distilled water. For every experiment, a new membrane sample was used. The concentrations were found using visible UV spectrophotometry for the dye solution at the maximum absorption wavelength. Adsorption tests were also performed. A membrane sample was installed, and 500 ppm of the dye solution was poured into the cell. Adsorption was followed versus time by measuring the decrease in absorbance of the solution [24].

### 2.3. Characterization of Nanocomposite Membranes

Morphologies of the nanocomposite membranes were characterized using Scanning Electron Microscopy (FEI, Quanta FEG 450 Netherlands). Energy-dispersive X-ray spectroscopy (EDX AMETEK USA) was used to study the elemental composition of the nanocomposite membrane.

Crystal structure X-ray diffraction spectroscopy was used to characterize nanomaterials. The chemical structure and crystallinity of nanocomposites were studied through X-ray diffraction (Model D/max 2250, Rigaku Corporation, and Japan) with Cu radiation, which was operated at 40 kV and 200 mA.

Fourier transform infrared spectroscopy was used to detect chemical compounds. FTIR spectrometer (Model Nicolet iS10, Thermo Fisher Scientific USA) was used with attenuated total reflection (ATR) accessory in transmission mode. The spectra were recorded in the range of 4000 to 500 cm^−1^ using 32 scans and a resolution of 4 cm^−1^. This technique depends on reflectance, absorbance and transmittance of light. The unique frequency molecules vibrate or rotate according to discrete energy levels. Generally, a molecule that exhibits IR adsorptions have a certain feature, namely an electric dipole moment of the molecule with a separation distance between positive and negative charges [25].

TGA analysis of elaborated virgin and aminated functionalized graphene oxide nanocomposite membranes was performed to study the thermal properties of the membrane. The range of heating of the sample was from 0 to 800 °C at 14 °C/min for 1 h by carrying nitrogen gas in simultaneous thermal analyzer (STA8000, PerkinElmer, Mount Berry, GA, USA) [26]. A ZWICK/ROELL Z 2.5 test unit was used to measure membranes’ mechanical properties using the procedure already described in the literature [27,28,29]. Membranes’ porosity was determined by a method reported in the literature [30] according to which the average membrane porosity can be calculated as the volume of the pores divided by the total volume of the membrane. Dried-out samples were weighed with an accurate measuring balance. Moreover, samples at that time were soaked in kerosene oil for 24 h and weighed again [23,31,32]. Generally, porosity εm was calculated by the given formula:(1)$$\varepsilon_{m}\left(\%\right)=\frac{\left[\frac{w_{1}-w_{2}}{D_{k}}\right]}{\left[\frac{w_{1}-w_{2}}{D_{k}}+\frac{w_{2}}{D_{pol}}\right]}*100$$
where w_1_ = weight of the wet membrane, w_2_ = weight of the dry membrane, Dk = density of kerosene oil (0.82 g/cm^3^) and Dpol = density of polymer: PVDF = 1.78 g/cm^3^. For every membrane, three measurements were taken, and average values and standard deviation were calculated.

Hydrophilicity or hydrophobicity of membranes can be determined through contact angle measurements. The measurement of contact angle is related to a three-phase equilibrium that occurs at the contact point between the liquid/liquid/liquid or solid/liquid/vapor interfaces [33]. If the involved phases are membrane/water/air, the contact angle (θ) of the membrane is a measure of its wettability, i.e., the capacity of water to be absorbed by the membrane’s surface [34]. The contact angle to water of the produced membranes was measured by the sessile drop method, which is a direct measurement of θ on a liquid drop deposited on the membrane surface. In particular, θ was determined by constructing a tangent to the profile at the point of contact between the drop and the membrane surface using an optical tensiometer. For each membrane type, specimens were cut into 20 mm × 1.5 mm strips and mounted on the sample holder. A 5 μL water droplet was deposited on the membrane with a microliter syringe. A light source was placed behind the sample, and the contact angle was measured with the tensiometer (Attension Theta Tensiometer Finland) [29].

Membranes’ performance was estimated by water permeability. Cross-flow disc holder, EPDM version 90 mm purchased from Sterlitech, working with an effective membrane area of 50 cm^2^, was used to determine the pure water permeability of the nanocomposite membranes. Prior to permeability measurement, membranes were soaked in ultra-pure water for 3 h. Membranes were initially compacted at 20 bar transmembrane pressure for 1 h. Permeability measurements were conducted at 25 °C and at constant flow rate (1 L/min), varying the pressure from 1 bar to 4 bar.
(2)$$J=\frac{V}{A.t}\ldots$$
where J is the water flux and V is the volume in liter of permeated water. A represents the area of membrane and t is the permeation time, whose units are h^−1^ [29].

## 3. Result and Discussion

### 3.1. Scanning Electron Microscopy

The thin-layer structure of the composite membrane (in terms of surface morphology, thin layer thickness and surface roughness) was strongly related to the separation, water flux and antifouling property of the NF membrane and could be characterized by scanning electron microscopy. SEM analysis of the graphene oxide and aminated graphene oxide is shown in Figure 3.

Scanning electron microscope gave information about the membrane surface, its composition, topography and texture through back-scattered secondary electrons, which generate signals to produce the images with good resolution. Additionally, the cross section of the membranes could also be determined by scanning electron microscopy. Surface morphology of aminated graphene oxide (AGO)-based PVDF membranes was also characterized by this technique, showing the presence of composite on the membrane surface, which confirms the interaction of amine, carboxyl, carbonyl and hydroxyl groups of AGO with the PVDF. However, their proper mechanism is not confirmed yet. All the composite membranes show the pattern of their morphology [35].

Figure 4A–F represents the SEM images of polyvinylidene fluoride (PVDF) membranes and 0.1% graphene oxide–based PVDF membranes. There is no particle embedded on the PVDF membranes, but there are particles embedded on graphene oxide–based polyvinylidene fluoride (PVDF) [36].

Scanning electron microscopy (SEM) images are shown in Figure 4A–F. All scanning electron microscope (SEM) images show the presence of aminated graphene oxide on the surface of membranes. M1 shows the absence of nanocomposite on the membrane surface. M2 shows the presence of nanocomposite on the membrane surface. Furthermore, other various compositions of AGO-PVDF membranes’ morphology show the pores present in the membrane and aminated graphene oxide particles dispersed on the membrane surface on AGO-1.0% membrane.

Figure 4 reveals pristine and AGO-doped PVDF nanocomposite membranes, exhibiting the dense skin layer and characteristic, a fingerlike porous sub-layer. It is evident that the pristine PVDF membrane has a narrow finger-like pattern. It is worth pointing out that these macrovoids appear to extend throughout this cross section by the incorporation of AGO nanoparticles and become wider at the bottom of membrane, resulting in an increase in water flux. The same findings were observed in the previously published work [23]. It is obvious that the relative diffusion rate and driving force between non-solvent and solvent are the characteristic parameters for structure of the membranes. Consequently, it is assumed that the nanoparticles have a better affinity to non-solvent, establishing elongated macrovoids and finger-like structures because of the superior exchange rate of the solvent/non-solvent.These significant variations in morphologies of membranes’ cross-sectional images are due to the improved de-mixing process and are in good agreement with previous research [23,29]. Moreover, there is no apparent accumulation of AGO nanoparticles in the membrane cross-section images, implying that the AGO nanoparticles were well distributed inside the membrane matrix.

### 3.2. X-ray Diffraction

As shown in Figure 5a, the peak at 11° corresponds to the graphene oxide as in our previous research [37], while there are sharp and broaden peaks of aminated graphene oxide at 2θ = 17.5° and 2θ = 26.5° as shown in Figure 5b. The broadened peaks of X-ray diffraction show that the structure of aminated graphene oxide is not consistent as natural graphite is. As chemical grafting forms strong bonding among graphene oxide and the attached molecules, and it can produce further faults on the graphene oxide surface, thus weakening the graphene oxide structure [38]. The structure of aminated graphene oxide is well maintained, and the lamella areas are reduced. The introduction of aminated graphene oxide in PVDF and DMAc solvent causes the peaks of XRD to broaden and divide, due to the decrease in peaks’ sharpness [39].

PVDF membranes exhibit crystalline peaks at 2θ = 18.4° and 20.8°, corresponding to the (020) and (110) reflections of the α phase, respectively [40]. Furthermore, the diffuse peak was shattered down into further minor peaks, and such peaks were made sharp by the addition of aminated graphene oxide, as shown in Figure 6. 

### 3.3. Fourier-Transform Infrared Spectroscopy

Fourier-transform infrared spectra (FTIR) of graphene oxide–based PVDF membranes with different composition of aminated graphene oxide are shown in Figure 7. All membranes indicate ranges of polyvinylidene fluoride and peaks at 1396 and 1175 cm^−1^, which might be recognized as the stretching and deformation vibrations of CH_2_ and the C–F stretching vibration [41]. The most noticeable feature of graphene oxide range is the adsorption peak at 3400 cm^−1^, which is related to the O–H stretching vibrations. The peaks at 2927 and 2850 cm^−1^ are due to the existence of the CH_2_ bond. The absorption peak at 1724 cm^−1^ is attributed to C=O carbonyl stretching, and the peak at 1624 cm^−1^ is due to C=C stretching as part of the phenol ring in the graphene oxide skeleton. The absorption peak at 1041 cm^−1^ is attributed to the stretching vibration of C–O groups [42]. The M1 membrane shows no peak for the N-H bond, whereas other membranes show the peak of the N-H bond at 1599 cm^−1^ to 1611 cm^−1^ [43].

### 3.4. Thermogravimetric Analysis

Figure 8 shows the TGA analysis of the elaborated virgin and aminated functionalized graphene oxide nanocomposite membranes. It is shown that adding AGO enhanced the thermal stability of membranes. The interaction among AGO and polymer enhanced the rigidity in polymer chain, which further enhanced the energy of breaking down the polymer chain [39,44]. However, further increase in AGO content led to the decrease in thermal stability, which might be due to the aggregation of composite. Moreover, the order of weight residual left at the end of analysis in line with the loading of aminated graphene oxide within polyvinylidene fluoride layer.

### 3.5. Tensile Strength

The tensile test was done to study the mechanical properties of membranes. Tensile strength of fabricated membranes and porosity are given in Figure 9. Tensile strength shows the mechanical properties of membrane, and our results revealed that mechanical strength of M4 and M5 membranes increases with the incorporation of aminated graphene oxide composites. This might be due to the strong interfacial bonding among the polymer chains and functional groups of composite, while the strength of other membranes decreases due to the increase in porosity [45]. The amine group itself helps to interact with the polymer and causes great compatibility between the composite and the polymer. However, the further addition of nanomaterial decreases the tensile strength, as in M6 membrane, while the thermal properties decrease [46,47].

### 3.6. Membrane Porosity and Contact Angle

Membrane porosity was determined by the gravimetric method described in previous work [31,32]. The membranes’ porosity was in the range of 87.71–90.70%. By looking at the reported values, it can be noticed that polymer and additive weight percent in the casting solution are the main factors affecting membrane porosity [29]. So, plotting the graph against tensile strength and porosity shows that by the addition of aminated graphene oxide, the porosity of PVDF membranes increases. The contact angle of membranes is commonly determined through sessile drop method, in which a goniometer instrument is used to measure the contact angle [48]. AGO-based PVDF membranes show the following trend in contact angle. The membrane with PVDF and solvent without the addition of aminated graphene oxide show a contact angle of 85.13°. With the addition of aminated graphene oxide, the contact angle decreases. This show that the membranes are hydrophilic. The trend of contact angle is shown in Figure 10. Due to the inherent hydrophilic property of PVDF and solvent and functionalized graphene oxide, all modified membranes possess an extremely low contact angle of water, which confirms their excellent hydrophilicity.

### 3.7. Water Permeability

Membrane performance was estimated by water flux and is presented in Table 2. Aminated graphene oxide (AGO) membranes showed an enhanced water permeability trend with the incorporation of AGO composites. Various factors are involved that enhanced water permeability of AGO membranes. Firstly, hydrophilicity of membranes was improved by the incorporation of AGO composites, which directly resulted in enhanced water flux. An increase in hydrophilicity led to an increase in water flux because of the attraction of water molecules towards membrane matrix and allowed them to pass through membranes. Secondly, by incorporation of nanomaterial, polymer chains may dislocate, leading to an increase in free volume or creating voids. Thirdly, the incorporation of aminated graphene oxide may have provided the pathways for water molecules to pass quickly through membranes. During fabrication, the increases in viscosity of the casting solution led to the decrease in the number of pores of membranes and the water flux of all other fabricated membranes.

Filtration tests were performed for each membrane with bovine serum albumin solution. The results are shown in Table 2. Protein molecules attached to the surface of the membrane and caused pore blocking as bovine serum albumin is hydrophobic in nature. The M5 membrane represents a higher value of BSA rejection than all other membranes, which shows that the existence of greater concentration of aminated graphene oxide means a more hydrophilic character (Figure 11). The BSA adsorption of the surface-modified, AGO-containing membrane was considerably reduced, indicating an enhanced surface hydrophilicity. Surface hydrophilicity and porous structure of the membrane are factors for improved membrane performance and limiting the protein adsorption, which indicates enhanced fouling resistance. This trend shows that with increasing concentration of aminated graphene oxide, the polymeric membrane solution becomes more viscous, which results in blockage of pores because of greater concentration of nanoparticles that results in aggregation of nanoparticles [49]. 

Methyl orange and methylene blue show maximum dye removal at 96.6% and 88.5%. For methyl orange at maximum concentration, rejection increases and then decreases by increasing the concentration of aminated graphene oxide. Similarly, methylene blue shows maximum rejection for M2 membrane, and thus on further increase of concentration of AGO, rejection decreases (Figure 12). Several coordinating functional groups present on GO surface facilitate the adsorption of dye during rejection. The interaction between positively charged nitrogen groups of methylene blue and methyl orange and the COOH and OH group of GO sheets facilitate the removal of dye molecules from water. Furthermore, methylene blue showed an opposite trend that might be due to the coagulation of composite at high concentration and effective surface area is not sufficient for methylene blue degradation [50].

### 3.8. Antibacterial Activity

To enhance the life time of a membrane, its protection from microorganism’s adherence is important. Thus, the antibacterial activity of membranes was tested to estimate their inhibition property. Following standard procedures, Gram-negative bacteria (*Escherichia coli*) and Gram-positive bacteria (*Staphylococcus aureus*) were cultivated in a Mueller Hinton agar plate at 37 °C, and the results are summarized in Figure 13. Test bacteria were grown on nutrient medium and conserved for estimation of antibacterial tests. Moreover, 200 mL of nutrient broth was taken in a separate flask and kept in autoclave at a temperature of 121 °C at 15 psi pressure for 20–30 min. Following the autoclave reaction, media were cooled at room temperature. After this, the formerly cultured strain of bacteria was inoculated in flask. The nanocomposite membranes with various compositions were placed in each flask. The incubation was carried out at 35 °C for 18 h. Determination of optical density was done at 600 nm by UV-Vis spectrophotometer, followed by incubation [50,51]. The antibacterial activity of *Escherichia coli* increased with an increase in graphene oxide concentration in the membrane. In the case of the water purification membrane, microorganisms existing in the water can adhere to the membrane surface and might block the pores, which results in the reduction of water flux. The significance of antibacterial membrane in water purification is that the surface of the antibacterial membrane hinders the growth of bacteria on the surface and inhibits the biofilm formation [52]. As compared to *E. coli*, *Staphylococcus aureus* showed less inhibition on polymeric membranes. Therefore, it has been shown that *E. coli* was more resistant to aminated graphene oxide polymeric membranes when concentration increased. When the GO concentrations were increased, bacteria cell results were more covered, effectively isolated from the environment and condemned to die. The bacteria cells, treated with aminated grapheme oxide GO, exhibited a significant reduction to their growth, as the concentration increased [53,54].

## 4. Conclusions

Aminated graphene oxide–modified polyvinylidene fluoride membranes with anti-biofouling, dyes (methylene blue and methylene orange) rejection and antibacterial properties were fabricated. The interaction between aminated graphene oxide PVDF membranes favors its better thermal and mechanical stability. Membranes show higher water permeability and more dye rejection due to the hydrophilic nature. Thus, these modified membranes proved to be efficient for water separation.

## Figures and Tables

**Figure 1 membranes-12-00224-f001:**
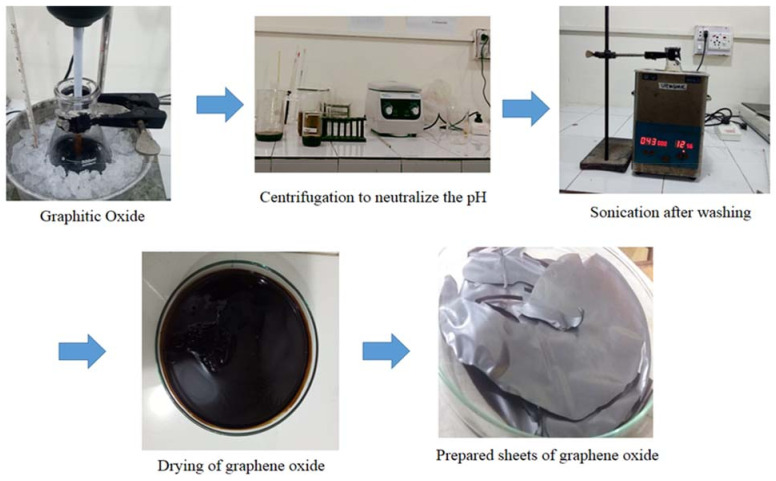
Schematic representation of the synthesis of graphene oxide.

**Figure 2 membranes-12-00224-f002:**
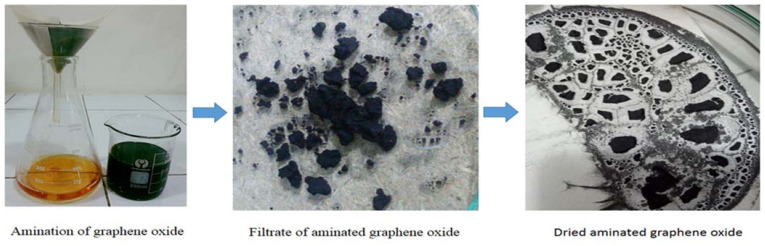
Schematic representation of the synthesis of aminated graphene oxide.

**Figure 3 membranes-12-00224-f003:**
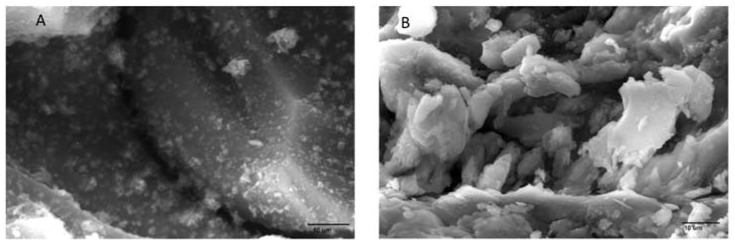
SEM images of aminated graphene oxide (**A**) and graphene oxide (**B**).

**Figure 4 membranes-12-00224-f004:**
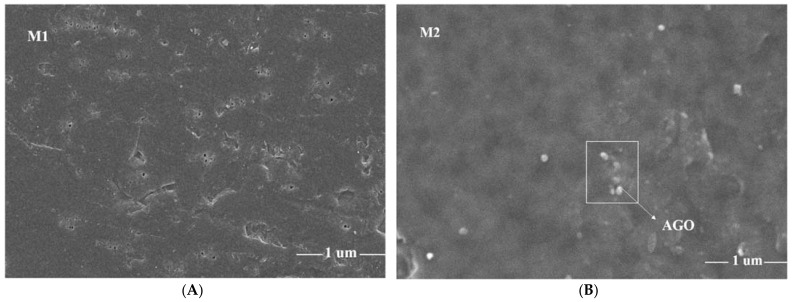
SEM images of M1 (**A**), M2 (**B**), M3 (**C**), M4 (**D**) M5, (**E**) and M6 (**F**) membranes. (**G**) Cross sectional SEM images of membranes M1(Pristine) and M6(1% AGO).

**Figure 5 membranes-12-00224-f005:**
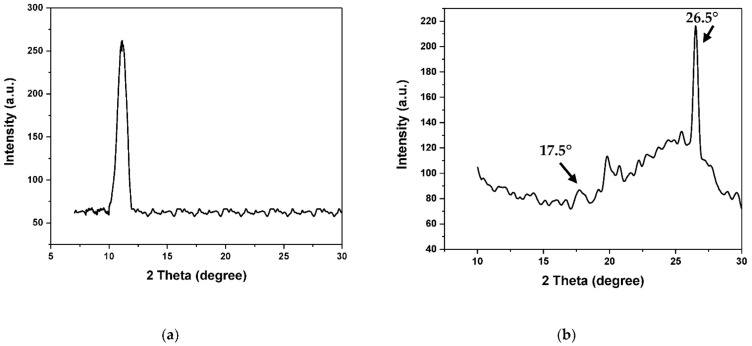
XRD analysis of graphene oxide (**a**) [2] and aminated graphene oxide (**b**).

**Figure 6 membranes-12-00224-f006:**
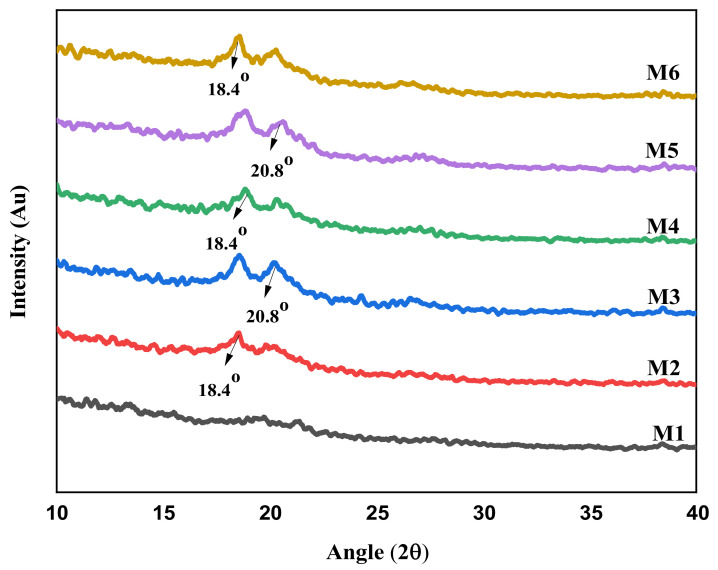
XRD analysis of aminated graphene oxide (AGO) based PVDF membranes.

**Figure 7 membranes-12-00224-f007:**
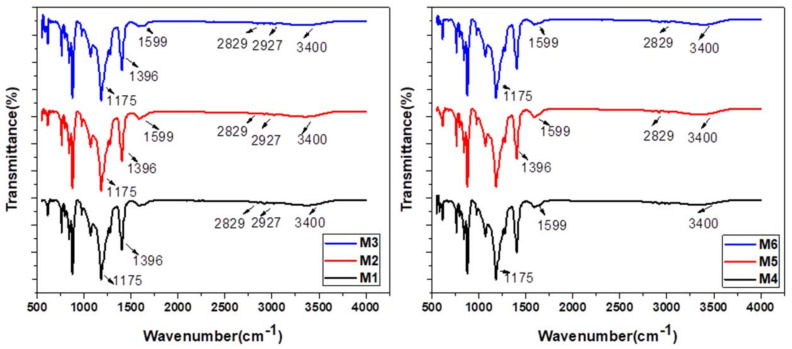
FTIR analysis of aminated graphene oxide (AGO)-based PVDF membranes.

**Figure 8 membranes-12-00224-f008:**
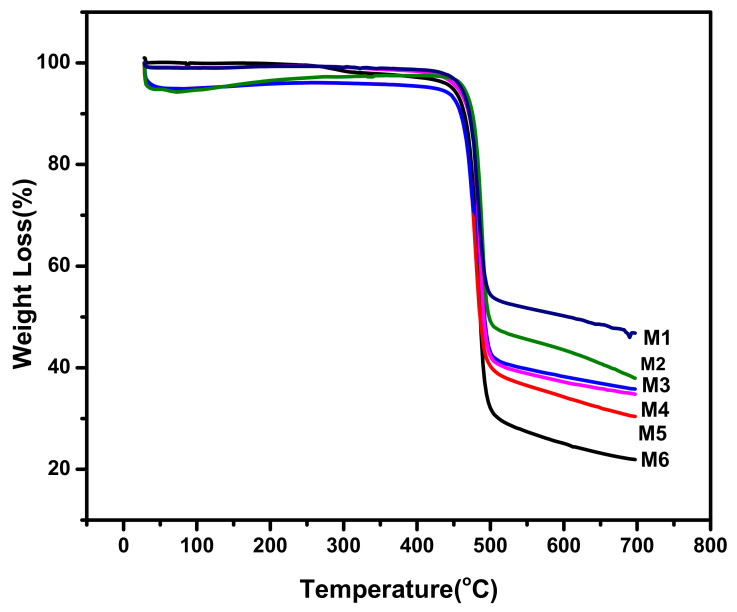
TGA graph of aminated graphene oxide–based PVDF membranes.

**Figure 9 membranes-12-00224-f009:**
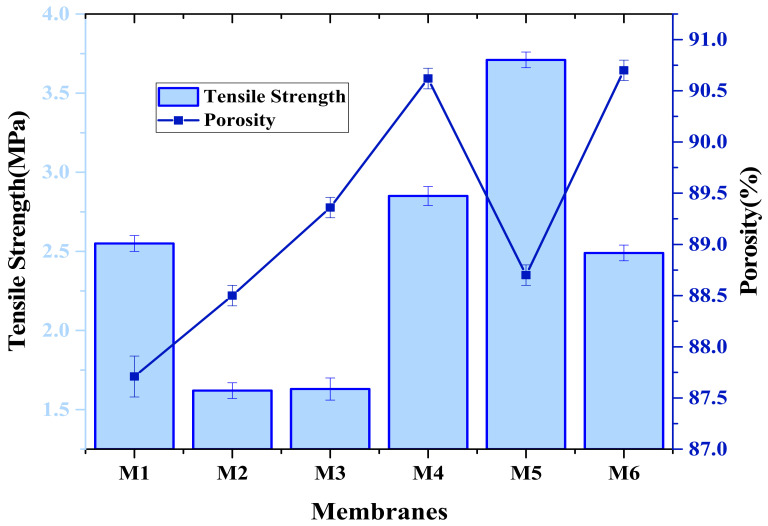
Mechanical properties vs. porosity of aminated graphene oxide based PVDF membranes.

**Figure 10 membranes-12-00224-f010:**
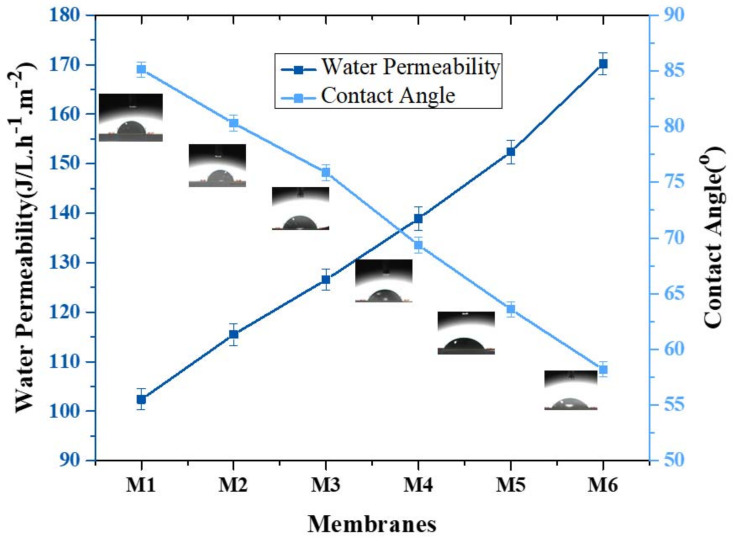
Trend of water permeability vs. contact angle of aminated graphene oxide–based PVDF membranes.

**Figure 11 membranes-12-00224-f011:**
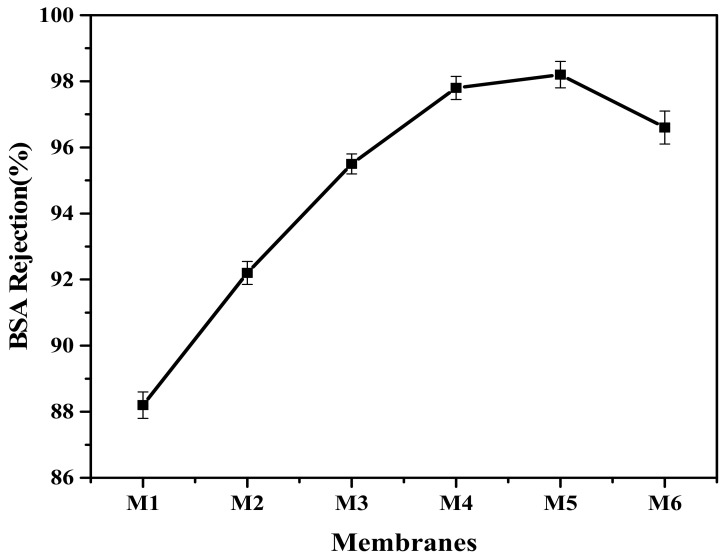
BSA rejection of aminated graphene oxide–based PVDF membranes.

**Figure 12 membranes-12-00224-f012:**
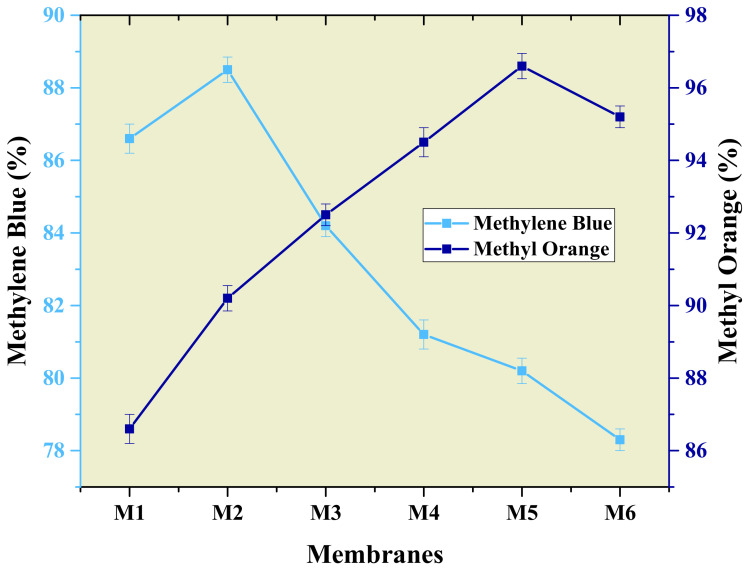
Trend of rejection of dyes through aminated graphene oxide–based PVDF membranes.

**Figure 13 membranes-12-00224-f013:**
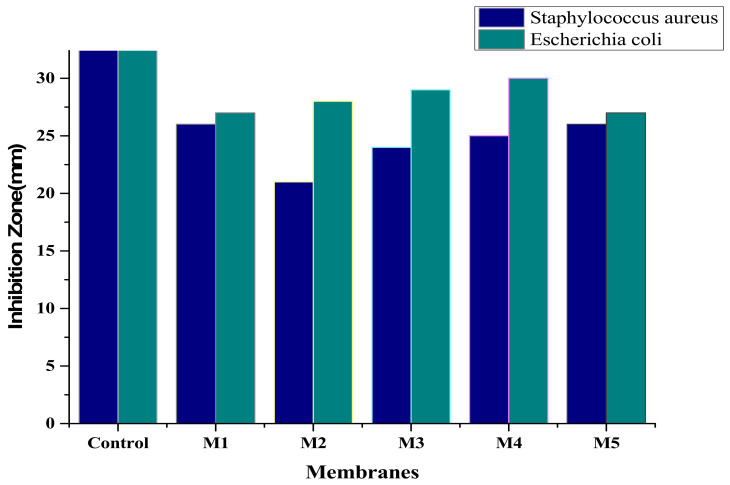
Inhibition zone of aminated graphene oxide–based PVDF membranes.

**Table 1 membranes-12-00224-t001:** Compositions of different fabricated nanocomposite membranes.

Membrane	PVDF (wt.%)	DMAc (wt.%)	AGO (wt.%)
M1	18	82	0
M2	18	81.8	0.2
M3	18	81.6	0.4
M4	18	81.4	0.6
M5	18	81.2	0.8
M6	18	81.0	1.0

**Table 2 membranes-12-00224-t002:** Water permeability, BSA rejection and dyes removal of AGO based PVDF membranes.

Membrane Samples	Water Permeability (J/L.h^−1^.m^−2^)	BSA Rejection (%)	Dye Rejection (Methyl Orange) (%)	Dye Rejection (Methylene Blue) (%)
M1	102.4	88.2	86.6	86.6
M2	115.5	92.2	90.2	88.5
M3	126.6	95.5	92.5	84.2
M4	138.9	97.8	94.5	81.2
M5	152.4	98.2	96.6	80.2
M6	170.2	96.6	95.2	78.3

## Data Availability

Not applicable.

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
