# Peer review of "Preparation of Polyvinylidene Fluoride Nano-Filtration Membranes Modified with Functionalized Graphene Oxide for Textile Dye Removal"

_membranes, 2022, doi:10.3390/membranes12020224_

Round 1

Reviewer 1 Report

The authors prepared the nanocomposite membrane for textile dyes removal. However, there are some major flaws and some missing elements as well as absence in the manuscript. In my opinion, the paper is not suitable to be published in Membranes. Some notable comments are as following.

  1. All acronyms must be defined at first time appearance in the manuscript.
  2. Please provide and illustrate the SEM or TEM images of pristine GO and AGO.
  3. SEM images of the cross sections for the each AGO-PVDF membranes should be provided. Please clearly discuss the dispersion.
  4. Please provide the XRD pattern of GO and compare it with AGO.
  5. The membrane thickness should be provided.
  6. Please clearly explain and prove the interaction between AGO and PVDF.
  7. Results and discussion: Figure 8. "Tensile strength shows the mechanical properties of membrane and our results revealed that mechanical strength of aminated graphene oxide membranes increases with the incorporation of aminated graphene oxide composites. " However, the tensile strength of membranes is not increased with the increasing amount of AGO. Especially, the tensile strength of M2 and M3 is lower than M1 (without AGO).
  8. What is the H5 membrane in Line 298?
  9. Please calculate the degree of amination. The degree of amination is possible that influences the properties of membranes and the capability of dyes removal.
  10. It is recommended that the author add negatively charged dyes for comparison and discussion.

Author Response

Comments and Suggestions for Authors

The authors prepared the nanocomposite membrane for textile dyes removal. However, there are some major flaws and some missing elements as well as absence in the manuscript. In my opinion, the paper is not suitable to be published in Membranes. Some notable comments are as following.

  1. All acronyms must be defined at first time appearance in the manuscript.

Response: Thanks for the comments. The changes are incorporated as per suggestions.

  1. Please provide and illustrate the SEM or TEM images of pristine GO and AGO.

Response: Thanks for the comments. The changes are incorporated as per suggestions for SEM and marked as figure 2. we don’t have TEM here at the institute and due to this COVID pandemic, the testing from other universities is not possible due to the unavailability of staff, closure of different testing facilities.

  1. SEM images of the cross sections for the each AGO-PVDF membranes should be provided. Please clearly discuss the dispersion.

Response: Figure 3(b) reveals pristine and AGO doped PVDF nanocomposite membranes, exhibiting the dense skin layer and characteristic, a fingerlike porous sub-layer. It is evident Pristine PVDF membrane has a narrow finger-like pattern. It is obvious worth to describe that these macrovoids appear to extend throughout this cross section by the incorporation of AGO nanoparticles and become wider at the bottom of membrane, resulting in increase of water flux. The same findings were observed in the previous work published [23]. It is obvious that the relative diffusion rate and driving force between non-solvent and solvent are the characteristic parameters for structure of the membranes. Consequently, it is assumed that the nanoparticles have a better affinity to non-solvent, establishing elongated macrovoids and finger-like structure because of the superior exchange rate of solvent/non-solvent [Ag paper]. These significant variations in morphologies of membrane’s cross-sectional images are due to the improved de-mixing process and are in good agreement with previous research [23,29]. Moreover, there is no apparent accumulation of AGO nanoparticles in the membrane cross-section images implying that the AGO nanoparticles were well distributed inside the membrane matrix

  1. Please provide the XRD pattern of GO and compare it with AGO.

Response: Thanks for the comments. The XRD data of the membranes is incorporated in the manuscript but unfortunately due to out of order of this instrument, this characterization is not possible at the moment. I hope you understand the situation.

  1. The membrane thickness should be provided.

Response: The thickness of the membrane is around 120 um for all the membranes.

  1. Please clearly explain and prove the interaction between AGO and PVDF.

Response: Thanks for the response, The detail discussion is incorporated accordingly in the manuscript.

  1. Results and discussion: Figure 8. "Tensile strength shows the mechanical properties of membrane and our results revealed that mechanical strength of aminated graphene oxide membranes increases with the incorporation of aminated graphene oxide composites. " However, the tensile strength of membranes is not increased with the increasing amount of AGO. Especially, the tensile strength of M2 and M3 is lower than M1 (without AGO).

Response: Tensile strength shows the mechanical properties of membrane and our results revealed that mechanical strength of M4 and M5 membranes increase with the incorporation of aminated graphene oxide composites. This might be due to the strong interfacial bonding among the polymer chains and functional groups of composites while the strength of other membranes decreases due to the increase in porosity [45]

  1. What is the H5 membrane in Line 298?

Response: Thanks for the comments, this is typo mistake and it was M5 instead of H5.

  1. Please calculate the degree of amination. The degree of amination is possible that influences the properties of membranes and the capability of dyes removal.
  2. It is recommended that the author add negatively charged dyes for comparison and discussion.

There is no literature available for Aminated GO based membranes for the removal of negatively charged dyes. We’ll do experiment in our next publication.

Reviewer 2 Report

Authors developed aminated graphene oxide typed membranes for dye removal. Some of their analysis on the characterization data are qualitative and inaccurate. Additional reliable reference literatures with reasonable analysis should be included in the revised version. In addition, the enhancements by adding aminated graphene oxides in dye rejections and antibacterial activity are just more or less than 10% only. Therefore, the mentioning the rejection percentage values in the abstract causes an overestimation. Besides, the antibacterial activity test would be meaningless unless the exact composition used in the test was known and any reference membrane is tested also for the proper comparison and evaluation. Comments and questions are the following.

1. It would be better if they mention what is responsible for the high wafter flux and membrane performance of aminated graphene oxide based PVDF membranes.

2. Figure 3: It is difficult to compare SEM images measured with different scales. It is strongly recommend to provide them with the same magnification.

3. Figure 3: It is not clear what is the aminated graphene oxide on the SEM images.

4. Figure 3 caption used the Wt% of AGO. If so, it is better to add the membrane numbers (M1 to M6) also.

5. Figure 4: They need to measure pure graphene oxide before the amination for comparison. They mentioned the XRD peaks are broaden, but it is acceptable only when the reference data of graphene oxide, not aminated, is available for comparison.

6. Figure 4: They need to assign the peaks at 13.2o and 22.5o with the XRD peaks for crystalline planes of graphite. What are other peaks at about 32o, 36.5o, 40o, 45o and 50o?

7. Figure 6: They need to assign the peaks 2978, 2927, 2850, and 1396 cm-1 with CH, CH2, or CH3 rather than C-H only, with reliable reference literatures also.

8. Figure 6: It is strongly recommended to have a separate figure for the region between 2500 and 4000 cm-1 with a narrow interval between the spectra. It would provide a better view on the peaks in the region.

9. Figure 6: It is not clear whether the peaks shift with a trend as the content of AGO is increased. If so, they need to interpret the reason.

10. Figure 7: M6 begins the weight loss at a lower temperature than M1, but interpreted that adding AGO enhance the thermal stability of membranes. Does it make sense?

11. Figure 8: Although authors concluded that tensile strength is increased with the incorporation of aminated graphene oxide composite, only M5 show 40% higher strength than M1. Others compositions made the strength similar (M4, M6) or even worse (M2, M3). Hence, their conclusions could be acceptable.

12. Figure 9: They need to provide the photos of contact angle measurements.

13. Figure 11: Methylene blue shows the opposite trend with membranes. What is the reason?

14. As for the antibacterial activity, they didn’t mention which membrane sample among M1-M6 was used, but having various composition. The performance results could be meaningless unless the exact composition is known and any reference membrane is tested also for proper comparison and evaluation.

Author Response

Comments and Suggestions for Authors

Authors developed aminated graphene oxide typed membranes for dye removal. Some of their analysis on the characterization data are qualitative and inaccurate. Additional reliable reference literatures with reasonable analysis should be included in the revised version. In addition, the enhancements by adding aminated graphene oxides in dye rejections and antibacterial activity are just more or less than 10% only. Therefore, the mentioning the rejection percentage values in the abstract causes an overestimation. Besides, the antibacterial activity test would be meaningless unless the exact composition used in the test was known and any reference membrane is tested also for the proper comparison and evaluation. Comments and questions are the following.

  1. It would be better if they mention what is responsible for the high wafter flux and membrane performance of aminated graphene oxide based PVDF membranes.

Response: Thanks for the comments. The detailed discussion is incorporated already in the manuscript and highlighted as well.

  1. Figure 3: It is difficult to compare SEM images measured with different scales. It is strongly recommend to provide them with the same magnification.

Response: Thanks for the comments and I do agree that the magnification should be same. The changes are incorporated accordingly.

  1. Figure 3: It is not clear what is the aminated graphene oxide on the SEM images.

Response: Thanks for the comments. The clear discussion as well as indication of presence of AGO in SEM micrograph.

  1. Figure 3 caption used the Wt% of AGO. If so, it is better to add the membrane numbers (M1 to M6) also.

Response: The changes are made accordingly as per suggestion of the reviewer.

  1. Figure 4: They need to measure pure graphene oxide before the amination for comparison. They mentioned the XRD peaks are broaden, but it is acceptable only when the reference data of graphene oxide, not aminated, is available for comparison.

Response: I do agree with reviewer, unfortunately the XRD is out of order and at the moment, the outer facility is also limited due to COVID-19.

  1. Figure 4: They need to assign the peaks at 13.2oand 22.5owith the XRD peaks for crystalline planes of graphite. What are other peaks at about 32o, 36.5o, 40o, 45o and 50o?

Response: Thanks for the comments. Other peaks at about 32o, 36.5o, 40o, 45o and 50o might be due to the presence of impurities. These peaks are not so sharp.

  1. Figure 6: They need to assign the peaks 2978, 2927, 2850, and 1396 cm-1with CH, CH2, or CH3rather than C-H only, with reliable reference literatures also.

Response:  The data presented in the FTIR discussion and Figure 6 is referred with reliable literature.

  1. Figure 6: It is strongly recommended to have a separate figure for the region between 2500 and 4000 cm-1with a narrow interval between the spectra. It would provide a better view on the peaks in the region.
  2. Figure 6: It is not clear whether the peaks shift with a trend as the content of AGO is increased. If so, they need to interpret the reason.

Response: Thanks for the comments and there is a peak shift but you cannot quantitively calculate the data and also its tricky to determine the overlapping peak as well in the same region.

  1. Figure 7: M6 begins the weight loss at a lower temperature than M1, but interpreted that adding AGO enhance the thermal stability of membranes. Does it make sense?

Response: Thank for the comments. The interaction among AGO and polymer enhanced the rigidity in polymer chain that further enhanced the energy of breaking down the polymer chain [39,44]. However, with further increase of AGO content lead to the decrease of thermal stability that might be due to the aggregation of composite. Moreover, the order of weight residual left at the end of analysis in line with the loading of aminated graphene oxide within polyvinylidene fluoride layer.

  1. Figure 8: Although authors concluded that tensile strength is increased with the incorporation of aminated graphene oxide composite, only M5 show 40% higher strength than M1. Others compositions made the strength similar (M4, M6) or even worse (M2, M3). Hence, their conclusions could be acceptable.

Response: Tensile strength shows the mechanical properties of membrane and our results revealed that mechanical strength of M4 and M5 membranes increase with the incorporation of aminated graphene oxide composites. This might be due to the strong interfacial bonding among the polymer chains and functional groups of composite while the strength of other membranes decreases due to the increase in porosity [45]. Amine group itself helps to interact with the polymer and causes great compatibility between the composite and polymer. While with the further addition of composite decreases the tensile strength as in M6 membrane as so thermal properties decrease

  1. Figure 9: They need to provide the photos of contact angle measurements.

Response: Thanks for the comments. The required information is not possible due to closure of lab facilities due to COVID-19.

  1. Figure 11: Methylene blue shows the opposite trend with membranes. What is the reason?

Response: Methyl orange and methylene blue shows maximum dye removal at 96.6 % and 88.5%. For methyl orange at maximum concentration rejection increases and then decreases by increasing the concentration of aminated graphene oxide. Similarly, methylene blue shows maximum rejection for M2 membrane and thus on further increase of concentration of AGO, rejection decreases (Figure 11). Several coordinating functional groups present on GO surface facilitate the adsorption of dye during rejection. The interaction between positively charged nitrogen groups of methylene blue and methyl orange and the COOH and OH group of GO sheets facilitate the removal of dye molecules from water [50]. Furthermore, methylene blue showed opposite trend that might be due to the coagulation of composite at high concentration and effective surface area is not sufficient for methylene blue degradation

  1. As for the antibacterial activity, they didn’t mention which membrane sample among M1-M6 was used but having various composition. The performance results could be meaningless unless the exact composition is known and any reference membrane is tested also for proper comparison and evaluation.

Response: thanks for the comments. The required information is included in the relevant section of the manuscript.

Reviewer 3 Report

Topic is well presented and organised in this paper. However, in order to clarify the novelty of the reported results, a table that shows the most important findings reported in the literature and your results must be added in the manuscript. 

Author Response

Comments and Suggestions for Authors

Topic is well presented and organised in this paper. However, in order to clarify the novelty of the reported results, a table that shows the most important findings reported in the literature and your results must be added in the manuscript. 

Response: Esteemed reviewer, thank you for your appreciated and valuable suggestion. We have incorporated your suggestion. Described the results section carefully in detail. Each part of the results section is improved according to your suggestion.

Round 2

Reviewer 1 Report

The authors have reasonably addressed most of my doubts, except for the followings:

  1. Please provide and illustrate the SEM or TEM images of pristine GO and AGO.
  2. Please calculate the degree of amination. The degree of amination is possible that influences the properties of membranes and the capability of dyes removal.
  3. It is recommended that the author add negatively charged dyes for comparison and discussion.

It is understandable that the COVID-situation had cost so many inconveniences to all of us. However, in the view of the reputable journal of "Membranes", I think that a certain level of novelty and technicality should be maintained in the published manuscript. Therefore, I would suggest Authors to further revise based on the above comments, which I think is critical to be qualified to match with the high standard of 'Membranes'.

Author Response

The authors have reasonably addressed most of my doubts, except for the followings:

  1. Please provide and illustrate the SEM or TEM images of pristine GO and AGO.

Response: Thanks for the comments. The SEM images of GO and AGO are included and also the detail discussion is incorporated. While We tried our level best for TEM images but currently not possible in nearby facilities.

  1. Please calculate the degree of amination. The degree of amination is possible that influences the properties of membranes and the capability of dyes removal.

Response: We have already addressed this point in previous comments that, we are unable to find any reference to calculate the degree of amination. If this is really needed it is requested to review to please guide us any technique or theoretical calculation method to address this comment.

  1. It is recommended that the author add negatively charged dyes for comparison and discussion.

Response: Thanks for the commentshe negative dye was not in the scope of the paper and we typically mentioned the specific dye for removal

It is understandable that the COVID-situation had cost so many inconveniences to all of us. However, in the view of the reputable journal of "Membranes", I think that a certain level of novelty and technicality should be maintained in the published manuscript. Therefore, I would suggest Authors to further revise based on the above comments, which I think is critical to be qualified to match with the high standard of 'Membranes'

Reviewer 2 Report

Authors replied reviewer’s comments and questions, but some of important supporting data (XRD, contact angle photo) are still missing. Even though it is due to Covid-19, they need to be included for supporting their data analysis in this reviewing process. In addition some of questions seem to be misunderstood. Comments and questions are the following.

1. Reviewer requested XRD data of pure graphene oxide before the amination for comparison, but authors replied their facility is limited due to Covid-19. It is one of important data for supporting their data analysis, so any results expectation or interpretation with skipping actual data wouldn’t be welcome. They said “limited”, not “impossible”. Nobody sure 100% until the data are obtained. It is highly recommended to provide authors enough time to add any missing data due to Covid-19, rather than skipping data collection. Hence, they may need to find any outside facility available for the XRD data.

2. If they considered impurities for other peaks at 32, 36.5, 40, 45, and 50o, they’re too much. Any efforts to remove them should be done prior to the measurements of water permeability and contact angles in Figure 9. It is strongly recommended to measure them after such a post-treatment.

3. Reviewer asked authors to separate the Figure 6, not overlapping peaks. They seem to be misunderstood. The IR peak intensities in the regions of 4000-2500 cm-1 are much weaker than them in 2000-500 cm-1. Hence, it is better to make two Figures. For example, Figure 6B displays from 2000-500 cm-1 with the current scale in Y axis. Figure 6A shows from 4000-2500 cm-1 with a different scale in Y axis. It would be easy if they decrease the interval between the spectra. It is common in displaying IR spectra. 

4. Figure 7: M3 get its weight, not loss in the region between 100 and 400 oC. What is the reason?

5. Figure 9: Authors already have the data of contact angle measurements, so reviewer asked to include the photos, they would have, supporting the angle values in this manuscript, not measuring them again. They replied the information is not possible as the facility is closed. If they didn’t copy them from the facility, another important supporting data are missing in this manuscript. 

Author Response

Authors replied reviewer’s comments and questions, but some of important supporting data (XRD, contact angle photo) are still missing. Even though it is due to Covid-19, they need to be included for supporting their data analysis in this reviewing process. In addition some of questions seem to be misunderstood. Comments and questions are the following.

  1. Reviewer requested XRD data of pure graphene oxide before the amination for comparison, but authors replied their facility is limited due to Covid-19. It is one of important data for supporting their data analysis, so any results expectation or interpretation with skipping actual data wouldn’t be welcome. They said “limited”, not “impossible”. Nobody sure 100% until the data are obtained. It is highly recommended to provide authors enough time to add any missing data due to Covid-19, rather than skipping data collection. Hence, they may need to find any outside facility available for the XRD data.

Response: The XRD data of GO and AGO is included in the manuscript as per suggestion of the reviewer.

  1. If they considered impurities for other peaks at 32, 36.5, 40, 45, and 50o, they’re too much. Any efforts to remove them should be done prior to the measurements of water permeability and contact angles in Figure 9. It is strongly recommended to measure them after such a post-treatment.

Response: The characterization of the samples were repeated for XRD and there is no impurities or additional peak. The figure is revised and included in the manuscript.

  1. Reviewer asked authors to separate the Figure 6, not overlapping peaks. They seem to be misunderstood. The IR peak intensities in the regions of 4000-2500 cm-1are much weaker than them in 2000-500 cm-1. Hence, it is better to make two Figures. For example, Figure 6B displays from 2000-500 cm-1 with the current scale in Y axis. Figure 6A shows from 4000-2500 cm-1 with a different scale in Y axis. It would be easy if they decrease the interval between the spectra. It is common in displaying IR spectra. 

Response: Thanks for the comments. The figure is splitted into two figures as per suggestion of the reviewers and the discussion is also revised.

  1. Figure 7: M3 get its weight, not loss in the region between 100 and 400 o What is the reason?

Response: Thank you very much for mentioning this point. We have carefully analysed the data and M3 was actually at wrong position while data interpretation. Now the figure is revised and there is no ambiguity now.

  1. Figure 9: Authors already have the data of contact angle measurements, so reviewer asked to include the photos, they would have, supporting the angle values in this manuscript, not measuring them again. They replied the information is not possible as the facility is closed. If they didn’t copy them from the facility, another important supporting data are missing in this manuscript. 

Response: Thank you very much for your valuable comments. We are unable to manage the photographs of contact angles. I am sure that reviewers understand that this will add on to manuscript but not technical comment.

Round 3

Reviewer 2 Report

Authors replied reviewer’s comments and questions as the required information is included in the relevant section of manuscript, but most of them are not clear. Where are their answers exactly? Which sentences? It is strongly recommended to mention which sentences are their answers. Further comments and questions are the following.

1. Page 10, Line 266: Authors mentioned “aminated graphene oxide particles aggregate on the membrane surface on AGO-1.0% membrane”, but the particles don’t look to be aggregated in Figure 3(a)-(F) for 1% AGO. Which particles are aggregated?

2. Figure 4: Authors mentioned the XRD peaks of aminated graphene oxide are weak and broaden at 17o and 22.5o in Figure 4(b), but they look still sharp. They need to add some Y ticks and numbers to compare their intensities with the peak at 11o of graphene oxide at 11o in Figure 4(a).

3. Figure 4(a): XRD data of graphene oxide looks strange as several patterns are repeating between 13o to 70o. It doesn’t seem to be real signal, compared to Figure 4(b). They need to measure it again to show any reproducible and reliable data.

4. Figure 4(a): They need to cite any reference literature for the peak of graphene oxide at 11o.

5. Figure 4(b): Authors interpreted the peak at 26.5o as very oxidative graphene. Do they have any reference literatures?

6. Figure 4(b): Authors interpreted the many peaks at above 30o as impurities, which are not small. If so, they need to figure out what kind of impurities they have on aminated graphene oxides. This is important because the impurities could contribute to water permeability and contact angle also in Figure 9. They need to provide more reliable explanation on this issue.

7. Figure 6. Reviewer asked about the assignments of the peaks at 2978, 2927, and 2850 cm-1, authors simply replied the FTIR data is referred with reliable literature. However, they still didn’t cite any reference for them. In addition, the term of C-H isn’t accurate yet, so further precise functional groups, either CH2 or CH3, symmetric or asymmetric, are required for the complete assignments. It is recommended to refer any IR reference book. 

8. Figure 9: Reviewer asked to provide the photos of contact angle measurements because they already have data, but authors replied they’re not possible due to closure of lab facilities due to covid-19. Absence of supporting data is quite difficult to be accepted, as especially Figure 9 is very important results in this manuscript. They’re supposed to save them when they measured. Covid-19 seems not to be a proper reason for any missing data, unless there is any journal policy for excuse. 

Author Response

Response to reviewer’s comments

Authors replied to reviewer’s comments and questions as the required information is included in the relevant section of manuscript, but most of them are not clear. Where are their answers exactly? Which sentences? It is strongly recommended to mention which sentences are their answers. Further comments and questions are the following.

  1. Page 10, Line 266: Authors mentioned “aminated graphene oxide particles aggregate on the membrane surface on AGO-1.0% membrane”, but the particles don’t look to be aggregated in Figure 3(a)-(F) for 1% AGO. Which particles are aggregated?

Response: Thanks for pointing out this typo error. The proper word id dispersed. We have mentioned in manuscript as well.

  1. Figure 4: Authors mentioned the XRD peaks of aminated graphene oxide are weak and broaden at 17oand 22.5o in Figure 4(b), but they look still sharp. They need to add some Y ticks and numbers to compare their intensities with the peak at 11o of graphene oxide at 11o in Figure 4(a).
  2. Figure 4(a): XRD data of graphene oxide looks strange as several patterns are repeating between 13oto 70o. It doesn’t seem to be real signal, compared to Figure 4(b). They need to measure it again to show any reproducible and reliable data.
  3. Figure 4(a): They need to cite any reference literature for the peak of graphene oxide at 11o.

Response: Thanks comments. For point 2,3,4. Please see below the response.

The data related to GO is already published in other journals and cited as well in current work. Secondly the peaks are also mentioned accordingly and same revised in manuscript.

  1. Figure 4(b): Authors interpreted the peak at 26.5oas very oxidative graphene. Do they have any reference literatures?

Response: Thanks comments. This particular line is removed from the manuscript.

  1. Figure 4(b): Authors interpreted the many peaks at above 30oas impurities, which are not small. If so, they need to figure out what kind of impurities they have on aminated graphene oxides. This is important because the impurities could contribute to water permeability and contact angle also in Figure 9. They need to provide more reliable explanation on this issue.

Response: Thanks comments. The figure is revised in order to avoid any ambiguity.

  1. Figure 6. Reviewer asked about the assignments of the peaks at 2978, 2927, and 2850 cm-1, authors simply replied the FTIR data is referred with reliable literature. However, they still didn’t cite any reference for them. In addition, the term of C-H isn’t accurate yet, so further precise functional groups, either CH2or CH3, symmetric or asymmetric, are required for the complete assignments. It is recommended to refer any IR reference book. 

Response: Thanks comments. The reference is cited accordingly to support the results.

  1. Figure 9: Reviewer asked to provide the photos of contact angle measurements because they already have data, but authors replied they’re not possible due to closure of lab facilities due to covid-19. Absence of supporting data is quite difficult to be accepted, as especially Figure 9 is very important results in this manuscript. They’re supposed to save them when they measured. Covid-19 seems not to be a proper reason for any missing data, unless there is any journal policy for excuse. 

Response: Thanks comments. The photos of the contact angles are mentioned in the figure.9 and I am sure now it satisfies the reviewer.